# Toward Learning Human-aligned Cross-domain Robust Models by Countering Misaligned Features

**Haohan Wang**[1]   **Zeyi Huang**[2]   **Hanlin Zhang**[1]   **Yong Jae Lee**[2]   **Eric P. Xing**[1,3,4]

[1]School of Computer Science, Carnegie Mellon University, Pittsburgh, PA, USA
[2]Department of Computer Sciences, University of Wisconsin-Madison, Madison, WI, USA
[3]Mohamed bin Zayed University of Artificial Intelligence, Abu Dhabi, United Arab Emirates
[4]Petuum, Inc., Pittsburgh, PA, USA

## Abstract

Machine learning has demonstrated remarkable prediction accuracy over *i.i.d* data, but the accuracy often drops when tested with data from another distribution. In this paper, we aim to offer another view of this problem in a perspective assuming the reason behind this accuracy drop is the reliance of models on the features that are not aligned well with how a data annotator considers similar across these two datasets. We refer to these features as misaligned features. We extend the conventional generalization error bound to a new one for this setup with the knowledge of how the misaligned features are associated with the label. Our analysis offers a set of techniques for this problem, and these techniques are naturally linked to many previous methods in robust machine learning literature. We also compared the empirical strength of these methods demonstrated the performance when these previous techniques are combined, with implementation available here.

## 1 INTRODUCTION

Machine learning, especially deep neural networks, has demonstrated remarkable empirical successes over various applications. The models even occasionally achieved results beyond human-level performances over benchmark datasets [*e.g.,* He et al., 2015]. However, whether it is desired for a model to outsmart human on benchmarks remains an open discussion in recent years: indeed, a model can create more application opportunities when it surpasses human-level performances, but the community also notices that the performance gain is sometimes due to model's exploitation of the features meaningless to a human, which may lead to unexpected performance drops when the models are tested with other datasets in practice that a human considers similar to the benchmark [Christian, 2020].

One of the most famous examples of the model's exploitation of non-human-aligned features is probably the usage of snow background in "husky vs. wolf" image classification [Ribeiro et al., 2016]. Briefly, when the model is trained to classify "husky vs wolf," it notices that wolf images usually have a snow background and learns to use the background features. This example is only one of many similar discussions concerning that the models are using features considered futile by humans [*e.g.,* Wang et al., 2019a, Sun et al., 2019], and, sometimes, the features used are not even perceptible to a human [Geirhos et al., 2019, Ilyas et al., 2019, Wang et al., 2020, Hermann and Kornblith, 2020]. The usage of these features might lead to a misalignment between the human and the models' understanding of the data, leading to a potential performance drop when the models are applied to other data that a human considers similar.

We illustrate this challenge with a toy example in Figure 1, where the model is trained on the source domain data to classify triangle vs. circle and tested on the target domain data with a different marginal distribution. However, the color coincides with the shape on the source domain. As a result, the model might learn either the shape function or the color one. The color function will not classify the target domain data correctly while the shape function can, but the empirical risk minimizer (ERM) cannot differentiate them and might learn either one, leading to potentially degraded performances during the test. As one might expect, whether shape or color is considered human-aligned is subjective depending on the task or the data and, in general, irrelevant to the statistical nature of the problem. Therefore, our remaining analysis will depend on such knowledge.

In this paper, we aim to formalize the above challenge to study the learning of human-aligned models. In particular, we derive a new generalization error bound when a model is trained on one distribution but tested on another one that human consider similar. As discussed previously, one potential challenge for this scenario is that the model may learn to

*Accepted for the 38th Conference on Uncertainty in Artificial Intelligence* (UAI 2022).

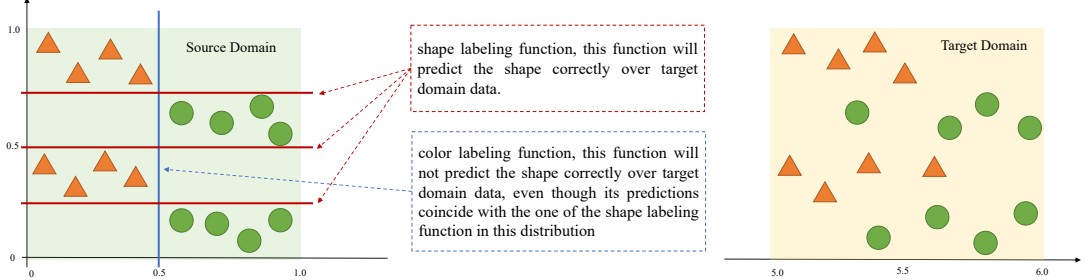

Figure 1: An illustration of the main problem focused in this paper.

use some features, which we refer to as *misaligned features*, that a human considers irrelevant. Corresponding to this challenge, our analysis will be built upon the knowledge of how misaligned features are associated with the label.

## 2 RELATED WORK

There is a recent proliferation of methods aiming to learn robust models by enforcing the models to disregard certain features. We consider these works direct precedents of our discussion because these features are usually defined when comparing the model's performances to a human's. For example, the texture or background of images is probably the most discussed misaligned features for image classification. We briefly discuss these works in two main strategies.

**Data Augmentation** With the knowledge of the misaligned features, the most effective solution is probably to augment the data by perturbing these misaligned features. Some recent examples of the perturbations used to train robust models include style transfer of images [Geirhos et al., 2019], naturalistic augmentation (color distortion, noise, and blur) of images [Hermann and Kornblith, 2020], other naturalistic augmentations (texture, rotation, contrast) of images [Wang et al., 2022], interpolation of images [Hendrycks et al., 2019a], syntactic transformations of sentences [Mahabadi et al., 2020], and across data domain [Shankar et al., 2018, Huang et al., 2020, Lee et al., 2021, Huang et al., 2022].

Further, as recent studies suggest that one reason for the adversarial vulnerability [Szegedy et al., 2013, Goodfellow et al., 2015] is the existence of imperceptible features correlated with the label [Ilyas et al., 2019, Wang et al., 2020], improving adversarial robustness may also be about countering the model's tendency toward learning these features. Currently, one of the most widely accepted methods to improve adversarial robustness is to augment the data along the training process to maximize the training loss by perturbing these features within predefined robustness constraints (*e.g.*, within $\ell_p$ norm ball) [Madry et al., 2018]. While this augmentation strategy is widely referred to as adversarial training, for the convenience of our discussion, we refer to it

as the worst-case data augmentation, following the naming conventions of [Fawzi et al., 2016].

**Regularizing Hypothesis Space** Another thread is to introduce inductive bias (*i.e.*, to regularize the hypothesis space) to force the model to discard misaligned features. To achieve this goal, one usually needs to first construct a side component to inform the main model about the misaligned features, and then to regularize the main model according to the side component. The construction of this side component usually relies on prior knowledge of what the misaligned features are. Then, methods can be built accordingly to counter the features such as the texture of images [Wang et al., 2019b, Bahng et al., 2019], the local patch of images [Wang et al., 2019a], label-associated keywords [He et al., 2019], label-associated text fragments [Mahabadi et al., 2020], and general easy-to-learn patterns of data [Nam et al., 2020].

In a broader scope, following the argument that one of the main challenges of domain adaptation is to counter the model's tendency in learning domain-specific features [*e.g.*, Ganin et al., 2016, Li et al., 2018], some methods contributing to domain adaption may have also progressed along the line of our interest. The most famous example is probably the domain adversarial neural network (DANN) [Ganin et al., 2016]. Inspired by the theory of domain adaptation [Ben-David et al., 2010], DANN trains the cross-domain generalizable neural network with the help of a side component specializing in classifying samples' domains. The subtle difference between this work and the ones mentioned previously is that this side component is not constructed with a special inductive bias but built as a simple network learning to classify domains with auxiliary annotations (domain IDs). DANN also inspires a family of methods forcing the model to learn auxiliary-annotation-invariant representations with a side component such as [Ghifary et al., 2016, Rozantsev et al., 2018, Motiian et al., 2017, Li et al., 2018, Carlucci et al., 2018].

**Relation to Previous Works** The above methods solve the same human-aligned learning problems with two different perspectives, but we notice the same central theme of

forcing the models to *not* learn something according to the prior knowledge of the data or the task. Although this central theme has been noticed by prior works such as [Wang et al., 2019b, Bahng et al., 2019, Mahabadi et al., 2020], we notice a lack of formal analysis from a task-agnostic viewpoint. Therefore, we continue to investigate whether we can contribute a principled understanding of this central theme, which serves as a connection of these methods and, potentially, a guideline for developing future methods. Also, we notice that many works along the domain adaptation development have rigorous statistical analysis [Ben-David et al., 2007, 2010, Mansour et al., 2009, Germain et al., 2016, Zhang et al., 2019, Dhouib et al., 2020], and these analyses mostly focus on the alignment of the distributions. Our study will complement these works by investigating through the perspective of misaligned features. The advantages and limitations of our perspective will also be discussed.

# 3 GENERALIZATION UNDERSTANDING OF HUMAN-ALIGNED ROBUST MODELS

**Roadmap** We study the generalization error bound of human-aligned robust model in this section. We will first set up the problem of studying the generalization of the model across two distributions, whose difference mainly lies in the fact that one distribution has another labelling function (namely, the misaligned labelling function) in addition to the one that is shared across both of these distributions (**A2**). Then, to help quantify the error bound, we need to define the active set (features used by the function) ($\mathcal{A}(f, \mathbf{x})$ in (3)), the difference between the two functions ($d(\theta, f, \mathbf{x})$ in (4)), and an additional term to quantify whether the model learns the function if the model can map the sample correctly ($r(\theta, \mathcal{A}(f, \mathbf{x}))$ in (5)). With these terms defined, we will show a formal result on the generalization error bound, which depends on how many training samples are predicted correctly when the model learns the mis-aligned samples in addition to the standard terms.

## 3.1 NOTATIONS & BACKGROUND

We consider a binary classification problem from feature space $\mathcal{X} \in \mathbb{R}^p$ to label space $\mathcal{Y} \in \{0, 1\}$. The distribution over $\mathcal{X}$ is denoted as $\mathbf{P}$. A *labeling function* $f : \mathcal{X} \to \mathcal{Y}$ is a function that maps the feature $\mathbf{x}$ to its label $y$. A *hypothesis* or *model* $\theta : \mathcal{X} \to \mathcal{Y}$ is also a function that maps the feature to the label. The difference in naming is only because we want to differentiate whether the function is a natural property of the space or distribution (thus called a labeling function) or a function to estimate (thus called a hypothesis or model). The hypothesis space is denoted as $\Theta$. We use dom to denote the domain (input space) of a function, thus $\text{dom}(\theta) = \mathcal{X}$.

This work studies the generalization error across two distributions, namely source and target distribution, denoted as $\mathbf{P}_s$ and $\mathbf{P}_t$, respectively. We are only interested when these two distributions are, considered by a human, similar but different: being similar means there exists a *human-aligned labeling function*, $f_h$, that maps any $\mathbf{x} \in \mathcal{X}$ to its label (thus the label $y := f_h(\mathbf{x})$); being different means there exists a *misaligned labeling function*, $f_m$, that for any $\mathbf{x} \sim \mathbf{P}_s$, $f_m(\mathbf{x}) = f_h(\mathbf{x})$. This "similar but different" property will be reiterated as an assumption (**A2**) later. We use $(\mathbf{x}, y)$ to denote a sample, and use $(\mathbf{X}, \mathbf{Y})_\mathbf{P}$ to denote a finite dataset if the features are from $\mathbf{P}$ (see detailed process from **A2**). We use $\epsilon_\mathbf{P}(\theta)$ to denote the expected risk of $\theta$ over distribution $\mathbf{P}$, and use $\widehat{\cdot}$ to denote the estimation of the term $\cdot$ (*e.g.*, the empirical risk is $\widehat{\epsilon}_\mathbf{P}(\widehat{\theta})$). We use $l(\cdot, \cdot)$ to denote a generic loss function.

For a dataset $(\mathbf{X}, \mathbf{Y})_\mathbf{P}$, if we train a model with

$$\widehat{\theta} = \arg\min_{\theta \in \Theta} \sum_{(\mathbf{x}, y) \in (\mathbf{X}, \mathbf{Y})_\mathbf{P}} l(\theta(\mathbf{x}), y), \qquad (1)$$

previous generalization study suggests that we can expect the error rate to be bounded as

$$\epsilon_\mathbf{P}(\widehat{\theta}) \leq \widehat{\epsilon}_\mathbf{P}(\widehat{\theta}) + \phi(|\Theta|, n, \delta), \qquad (2)$$

where $\epsilon_\mathbf{P}(\widehat{\theta})$ and $\widehat{\epsilon}_\mathbf{P}(\widehat{\theta})$ respectively are

$$\epsilon_\mathbf{P}(\widehat{\theta}) = \mathbb{E}_{\mathbf{x} \sim \mathbf{P}} |\widehat{\theta}(\mathbf{x}) - y| = \mathbb{E}_{\mathbf{x} \sim \mathbf{P}} |\widehat{\theta}(\mathbf{x}) - f_h(\mathbf{x})|$$

and

$$\widehat{\epsilon}_\mathbf{P}(\widehat{\theta}) = \frac{1}{n} \sum_{(\mathbf{x}, y) \in (\mathbf{X}, \mathbf{Y})_\mathbf{P}} |\widehat{\theta}(\mathbf{x}) - y|,$$

and $\phi(|\Theta|, n, \delta)$ is a function of hypothesis space $|\Theta|$, number of samples $n$, and the probability when the bound holds $\delta$. This paper expands the discussion with this generic form that can relate to several discussions, each with its own assumptions. We refer to these assumptions as **A1**.

**A1**: basic assumptions needed to derived (2), for example,

  – when **A1** is "$\Theta$ is finite, $l(\cdot, \cdot)$ is a zero-one loss, samples are *i.i.d*", $\phi(|\Theta|, n, \delta) = \sqrt{(\log(|\Theta|) + \log(1/\delta))/2n}$
  – when **A1** is "samples are *i.i.d*", $\phi(|\Theta|, n, \delta) = 2\mathcal{R}(\mathcal{L}) + \sqrt{(\log 1/\delta)/2n}$, where $\mathcal{R}(\mathcal{L})$ stands for Rademacher complexity and $\mathcal{L} = \{l_\theta \mid \theta \in \Theta\}$, where $l_\theta$ is the loss function corresponding to $\theta$.

For more information, we refer interested readers to relevant textbooks such as [Bousquet et al., 2003] for formal and intuitive discussions.

## 3.2 GENERALIZATION ERROR BOUND OF HUMAN-ALIGNED ROBUST MODELS

Formally, we state the challenge of our human-aligned robust learning problem as the assumption:

**A2**: **Existence of Misaligned Features:** For any $\mathbf{x} \in \mathcal{X}$, $y := f_h(\mathbf{x})$. We also have a $f_m$ that is different from $f_h$, and for $\mathbf{x} \sim \mathbf{P}_s$, $f_h(\mathbf{x}) = f_m(\mathbf{x})$.

Thus, the existence of $f_m$ is a key challenge for the small empirical risk over $\mathbf{P}_s$ to be generalized to $\mathbf{P}_t$, because $\theta$ that learns either $f_h$ or $f_m$ will lead to small source error, but only $\theta$ that learns $f_h$ will lead to small target error. Note that $f_m$ may not exist for an arbitrary $\mathbf{P}_s$. In other words, **A2** can be interpreted to ensure the a property of $\mathbf{P}_s$ so that $f_m$, while being different from $f_h$, exists for any $\mathbf{x} \sim \mathbf{P}_s$.

In this problem, $f_m$ and $f_h$ are not the same despite $f_m(\mathbf{x}) = f_h(\mathbf{x})$ for any $\mathbf{x} \sim \mathbf{P}_s$, and we focus on the case where the differences lie in the features they use. To describe this difference, we introduce the notation $\mathcal{A}(\cdot, \cdot)$, which denotes a set parametrized by the labeling function and the sample, to describe the *active set* of features used by the labeling function. By *active set*, we refer to the minimum set of features that a labeling function requires to map a sample to its label. Formally, we define

$$\mathcal{A}(f, \mathbf{x}) = \{i | \widehat{\mathbf{z}}_i = \mathbf{x}_i\}, \quad \text{where,}$$
$$\widehat{\mathbf{z}} = \underset{\mathbf{z} \in \text{dom}(f), f(\mathbf{z}) = f(\mathbf{x})}{\arg\min} |\{i | \mathbf{z}_i = \mathbf{x}_i\}|, \quad (3)$$

and $|\cdot|$ measures the cardinality. Intuitively, $\mathcal{A}(f, \mathbf{x})$ indexes the features $f$ uses to predict $\mathbf{x}$. Although $f_m(\mathbf{x}) = f_h(\mathbf{x})$, $\mathcal{A}(f_m, \mathbf{x})$ and $\mathcal{A}(f_h, \mathbf{x})$ can be different. $\mathcal{A}(f_m, \mathbf{x})$ is the *misaligned features* following our definition.

Further, we define a function difference given a sample as

$$d(\theta, f, \mathbf{x}) = \underset{\mathbf{z} \in \text{dom}(f) : \mathbf{z}_{\mathcal{A}(f, \mathbf{x})} = \mathbf{x}_{\mathcal{A}(f, \mathbf{x})}}{\max} |\theta(\mathbf{z}) - f(\mathbf{z})|, \quad (4)$$

where $\mathbf{x}_{\mathcal{A}(f, \mathbf{x})}$ denotes the features of $\mathbf{x}$ indexed by $\mathcal{A}(f, \mathbf{x})$. In other words, the distance describes: given a sample $\mathbf{x}$, the maximum disagreement of the two functions $\theta$ and $f$ for all the other data $\mathbf{z} \in \mathcal{X}$ with a constraint that the features indexed by $\mathcal{A}(f, \mathbf{x})$ are the same as those of $\mathbf{x}$. Notice that this difference is not symmetric, as the active set is determined by the second function. By definition, we have $d(\theta, f, \mathbf{x}) \geq |\theta(\mathbf{x}) - f(\mathbf{x})|$.

Also, please notice that when we use expressions such as $\mathbf{z}_{\mathcal{A}(f, \mathbf{x})} = \mathbf{x}_{\mathcal{A}(f, \mathbf{x})}$, we imply that $\mathcal{A}(f, \mathbf{x})$ is the same in both LHS and RHS. Under this premise of the notation, whether (3) has a unique solution or not will not affect our main conclusion.

In addition, one may notice the connection between $\mathcal{A}(f, \mathbf{x})$ and the minimum sufficient explanation discussed previously [*e.g.,* Camburu et al., 2020, Yoon et al., 2019, Carter et al., 2019, Ribeiro et al., 2018]. While $\mathcal{A}(f, \mathbf{x})$ is conceptually the same as the minimum set of features for a model to predict, we define it mathematically different.

To continue, we introduce the following assumption:

**A3**: **Realized Hypothesis:** Given a large enough hypothesis space $\Theta$, for any sample $(\mathbf{x}, y)$, for any $\theta \in \Theta$, which is not a constant mapping, if $\theta(\mathbf{x}) = y$, then $d(\theta, f_h, \mathbf{x}) d(\theta, f_m, \mathbf{x}) = 0$

Intuitively, **A3** assumes $\theta$ at least learns one labeling function for the sample $\mathbf{x}$ if $\theta$ can map the $\mathbf{x}$ correctly.

Finally, to describe how $\theta$ depends on the active set of $f$, we introduce the term

$$r(\theta, \mathcal{A}(f, \mathbf{x})) = \underset{\mathbf{z}_{\mathcal{A}(f, \mathbf{x})} \in \text{dom}(f)_{\mathcal{A}(f, \mathbf{x})}}{\max} |\theta(\mathbf{z}) - y|, \quad (5)$$

where $\mathbf{z}_{\mathcal{A}(f, \mathbf{x})} \in \text{dom}(f)_{\mathcal{A}(f, \mathbf{x})}$ denotes that the features of $\mathbf{z}$ indexed by $\mathcal{A}(f, \mathbf{x})$ are searched in the input space $\text{dom}(f)$. Notice that $r(\theta, \mathcal{A}(f, \mathbf{x})) = 1$ alone does not mean $\theta$ depends on the active set of $f$; it only means so when we also have $\theta(\mathbf{x}) = y$ (see the formal discussion in Lemma **??**). In other words, $r(\theta, \mathcal{A}(f, \mathbf{x})) = 1$ alone may not have an intuitive meaning, but given $\theta(\mathbf{x}) = y$, $r(\theta, \mathcal{A}(f, \mathbf{x})) = 1$ intuitively means $\theta$ learns $f$.

With all above, we can extend the conventional generalization error bound with a new term as follows:

**Theorem 3.1.** *With Assumptions A1-A3, $l(\cdot, \cdot)$ is a zero-one loss, with probability as least $1 - \delta$, we have*

$$\epsilon_{\mathbf{P}_t}(\theta) \leq \widehat{\epsilon}_{\mathbf{P}_s}(\theta) + c(\theta) + \phi(|\Theta|, n, \delta) \quad (6)$$

*where*

$$c(\theta) = \frac{1}{n} \sum_{(\mathbf{x}, y) \in (\mathbf{X}, \mathbf{Y})_{\mathbf{P}_s}} \mathbb{I}[\theta(\mathbf{x}) = y] r(\theta, \mathcal{A}(f_m, \mathbf{x})).$$

$\mathbb{I}[\cdot]$ is a function that returns 1 if the condition $\cdot$ holds and 0 otherwise. As $\theta$ may learn $f_m$, $\widehat{\epsilon}_{\mathbf{P}_s}(\theta)$ is not representative of $\epsilon_{\mathbf{P}_t}(\theta)$; thus, we introduce $c(\theta)$ to account for the discrepancy. Intuitively, $c(\theta)$ quantifies the samples that are correctly predicted, but only because the $\theta$ learns $f_m$ for that sample. $c(\theta)$ depends on the knowledge of $f_m$.

We name Theorem 3.1 *the curse of universal approximation* to highlight the fact that the existence of $f_m$ is not always obvious, but the models can usually learn it nonetheless [Wang et al., 2020] . Even in a well-curated dataset that does not seemingly have misaligned features, modern models might still use some features not understood by human. This argument may also align with recent discussions suggesting that reducing the model complexity can improve cross-domain generalization [Chuang et al., 2020].

## 3.3 IN COMPARISON TO THE VIEW OF DOMAIN ADAPTATION

We continue to compare Theorem 3.1 with understandings of domain adaptation. Conveniently, several domain adaptation analyses [Ben-David et al., 2007, 2010, Mansour et al.,

2009, Germain et al., 2016, Zhang et al., 2019, Dhouib et al., 2020] can be sketched in the following form:

$$\epsilon_{\mathbf{P}_t}(\theta) \leq \hat{\epsilon}_{\mathbf{P}_s}(\theta) + D_\Theta(\mathbf{P}_s, \mathbf{P}_t) + \lambda + \phi'(|\Theta|, n, \delta) \quad (7)$$

where $D_\Theta(\mathbf{P}_s, \mathbf{P}_t)$ quantifies the differences between the two distributions; $\lambda$ describes the nature of the problem and usually involves non-estimable terms about the problem.

For example, Ben-David et al. [2010] formalized the difference as $\Theta$-divergence, and described the corresponding empirical term as (with $\Theta \Delta \Theta$ denoting the set of disagreement between two hypotheses in $\Theta$):

$$D_\Theta(\mathbf{P}_s, \mathbf{P}_t) = 1 - \min_{\theta \in \Theta \Delta \Theta} \left( \frac{1}{n} \sum_{\mathbf{x}:\theta(\mathbf{x})=0} \mathbb{I}[\mathbf{x} \in (\mathbf{X}, \mathbf{Y})_{\mathbf{P}_s}] \right.$$
$$\left. + \frac{1}{n} \sum_{\mathbf{x}:\theta(\mathbf{x})=1} \mathbb{I}[\mathbf{x} \in (\mathbf{X}, \mathbf{Y})_{\mathbf{P}_t}] \right).$$
$$(8)$$

Also, Ben-David et al. [2010] formalized $\lambda = \epsilon_{\mathbf{P}_t}(\theta^\star) + \epsilon_{\mathbf{P}_s}(\theta^\star)$, where $\theta^\star = \arg\min_{\theta \in \Theta} \epsilon_{\mathbf{P}_t}(\theta) + \epsilon_{\mathbf{P}_s}(\theta)$,

In our discussion, as we assume the $f_h$ applies to any $\mathbf{x} \in \mathcal{X}$ (according to **A2**), $\lambda = 0$ as long as the hypothesis space is large enough. Therefore, the comparison mainly lies in comparing $c(\theta)$ and $D_\Theta(\mathbf{P}_s, \mathbf{P}_t)$.

To compare them, we need an extra assumption:

**A4**: **Sufficiency of Training Samples** for the two finite datasets in the study, *i.e.*, $(\mathbf{X}, \mathbf{Y})_{\mathbf{P}_s}$ and $(\mathbf{X}, \mathbf{Y})_{\mathbf{P}_t}$, for any $\mathbf{x} \in (\mathbf{X}, \mathbf{Y})_{\mathbf{P}_t}$, there exists one or many $\mathbf{z} \in (\mathbf{X}, \mathbf{Y})_{\mathbf{P}_s}$ such that

$$\mathbf{x} \in \{\mathbf{x}' | \mathbf{x}' \in \mathcal{X} \text{ and } \mathbf{x}'_{\mathcal{A}(f_h, \mathbf{z})} = \mathbf{z}_{\mathcal{A}(f_h, \mathbf{z})}\} \quad (9)$$

**A4** intuitively means the finite training dataset needs to be diverse enough to describe the concept that needs to be learned. For example, imagine building a classifier to classify mammals *vs.* fishes from the distribution of photos to that of sketches, we cannot expect the classifier to do anything good on dolphins if dolphins only appear in the test sketch dataset. **A4** intuitively regulates that if dolphins will appear in the test sketch dataset, they must also appear in the training dataset.

Now, in comparison to [Ben-David et al., 2010], we have

**Theorem 3.2.** *With Assumptions A2-A4, and if $1 - f_h \in \Theta$, we have*

$$c(\theta) \leq D_\Theta(\mathbf{P}_s, \mathbf{P}_t)$$
$$+ \frac{1}{n} \sum_{(\mathbf{x}, y) \in (\mathbf{X}, \mathbf{Y})_{\mathbf{P}_t}} \mathbb{I}[\theta(\mathbf{x}) = y] r(\theta, \mathcal{A}(f_m, \mathbf{x}))$$
$$(10)$$

*where*

$$c(\theta) = \frac{1}{n} \sum_{(\mathbf{x}, y) \in (\mathbf{X}, \mathbf{Y})_{\mathbf{P}_s}} \mathbb{I}[\theta(\mathbf{x}) = y] r(\theta, \mathcal{A}(f_m, \mathbf{x}))$$

*and $D_\Theta(\mathbf{P}_s, \mathbf{P}_t)$ is defined as in* (8).

$q(\theta) := \frac{1}{n} \sum_{(\mathbf{x}, y) \in (\mathbf{X}, \mathbf{Y})_{\mathbf{P}_t}} \mathbb{I}[\theta(\mathbf{x}) = y] r(\theta, \mathcal{A}(f_m, \mathbf{x}))$, which intuitively means that if $\theta$ learns $f_m$, how many samples $\theta$ can coincidentally predict correctly over the finite target set used to estimate $D_\Theta(\mathbf{P}_s, \mathbf{P}_t)$. For sanity check, if we replace $(\mathbf{X}, \mathbf{Y})_{\mathbf{P}_t}$ with $(\mathbf{X}, \mathbf{Y})_{\mathbf{P}_s}$, $D_\Theta(\mathbf{P}_s, \mathbf{P}_t)$ will be evaluated at 0 as it cannot differentiate two identical datasets, and $q(\theta)$ will be the same as $c(\theta)$. On the other hand, if no samples from $(\mathbf{X}, \mathbf{Y})_{\mathbf{P}_t}$ can be mapped correctly with $f_m$ (coincidentally), $q(\theta) = 0$ and $c(\theta)$ will be a lower bound of $D_\Theta(\mathbf{P}_s, \mathbf{P}_t)$.

The value of Theorem 3.2 lies in the fact that for an arbitrary target dataset $(\mathbf{X}, \mathbf{Y})_{\mathbf{P}_t}$, no samples out of which can be predicted correctly by learning $f_m$ (a situation likely to occur for arbitrary datasets since $f_m$ is unlikely to be shared across the source dataset and any arbitrary target dataset), $c(\theta)$ will always be a lower bound of $D_\Theta(\mathbf{P}_s, \mathbf{P}_t)$.

Further, when Assumption **A4** does not hold, we are unable to derive a clear relationship between $c(\theta)$ and $D_\Theta(\mathbf{P}_s, \mathbf{P}_t)$. The difference is mainly raised as a matter of fact that, intuitively, we are only interested in the problems that are "solvable" (**A4**, *i.e.*, hypothesis that used to reduce the test error in target distribution can be learned from the finite training samples) but "hard to solve" (**A2**, *i.e.*, another labeling function, namely $f_m$, exists for features sampled from the source distribution only), while $D_\Theta(\mathbf{P}_s, \mathbf{P}_t)$ estimates the divergence of two arbitrary distributions.

## 3.4 ESTIMATION OF THE DISCREPANCY

The estimation of $c(\theta)$ mainly involves two challenges: the requirement of the knowledge of $f_m$ and the computational cost to search over the entire space $\mathcal{X}$.

The first challenge is unavoidable by definition because the human-aligned learning has to be built upon the prior knowledge of what labeling function a human considers similar (what $f_h$ is) or its opposite (what $f_m$ is). Fortunately, as discussed in Section 2, the methods are usually developed with prior knowledge of what the misaligned features are, suggesting that we may often directly have the knowledge.

The second challenge is about the computational cost to search, and the community has several techniques to help reduce the burden. For example, the search can be terminated once $r(\theta, \mathcal{A}(f_m, \mathbf{x}))$ is evaluated as 1 (*i.e.*, once we find a perturbation of misaligned features that alters the prediction). This procedure is similar to how adversarial attack [Goodfellow et al., 2015] is used to evaluate the robustness of models. To further reduce the computational cost, one can also generate out-of-domain data by perturbing misaligned features beforehand and use these fixed data to test models. Using fixed data to evaluate might not be as accurate as using a search process, but sometimes, it can be good enough

to reveal some interesting properties of the models [Jo and Bengio, 2017, Geirhos et al., 2019, Wang et al., 2020].

# 4 METHODS TO LEARN HUMAN-ALIGNED ROBUST MODELS

We continue to study how our analytical results above can lead to practical methods to learn human-aligned robust models. We first show that our discussion can naturally connect to existing methods for robust machine learning discussed in Section 2.

Theorem 3.1 suggests that training a human-aligned robust model amounts to training for small $c(\theta)$ and small empirical error (*i.e.*, $\widehat{\epsilon}_{\mathbf{P}_s}(\theta)$).

## 4.1 WORST-CASE TRAINING

To simplify the notation, we define $\mathcal{Q}(\mathbf{x}) := \{\mathbf{x}_{\mathcal{A}(f_m, \mathbf{x})} \in \text{dom}(f_m)_{\mathcal{A}(f_m, \mathbf{x})}\}$. We can consider the upper bound of $c(\theta)$

$$
\begin{aligned}
c(\theta) \leq & \frac{1}{n} \sum_{(\mathbf{x}, y) \in (\mathbf{X}, \mathbf{Y})} r(\theta, \mathcal{A}(f_m, \mathbf{x})) \\
= & \frac{1}{n} \sum_{(\mathbf{x}, y) \in (\mathbf{X}, \mathbf{Y})} \max_{\mathbf{z} \in \mathcal{Q}(\mathbf{x})} |\theta(\mathbf{z}) - y|,
\end{aligned} \tag{11}
$$

which intuitively means that instead of $c(\theta)$ that studies only the correct predictions because $\theta$ learns $f_m$, now we study any predictions because $\theta$ learns $f_m$.

Further, as

$$
|\theta(\mathbf{x}) - y| \leq \max_{\mathbf{z} \in \mathcal{Q}(\mathbf{x})} |\theta(\mathbf{z}) - y|,
$$

a model with minimum (11) naturally means the model will have a minimum empirical loss. Therefore, we can train for a small (11), which likely leads to the model with a small empirical loss. Therefore, after we replace $|\theta(\mathbf{x}) - y|$ with a generic loss term $\ell(\theta(\mathbf{x}), y)$, we can directly train a model with

$$
\min_{\theta \in \Theta} \frac{1}{n} \sum_{(\mathbf{x}, y) \in (\mathbf{X}, \mathbf{Y})} \max_{\mathbf{z} \in \mathcal{Q}(\mathbf{x})} \ell(\theta(\mathbf{z}), y) \tag{12}
$$

to get a model with small $c(\theta)$ and small empirical error.

The above method is to augment the data by perturbing the misaligned features to maximize the training loss and solve the optimization problem with the augmented data. This method is the worst-case data augmentation method [Fawzi et al., 2016] we discussed previously, and is also closely connected to one of the most widely accepted methods for the adversarial robust problem, namely the adversarial training [Madry et al., 2018].

While the above result shows that a method for learning human-aligned robust models is in mathematical connection to the worst-case data augmentation, in practice, a general application of this method will require some additional assumptions. The detailed discussions of these are in the appendix.

We continue from the RHS of (11) to discuss another reformulation by reweighting sample losses for optimization, which leads to:

$$
\frac{1}{n} \sum_{(\mathbf{x}, y) \in (\mathbf{X}, \mathbf{Y})} \max_{\mathbf{z} \in \mathcal{Q}(\mathbf{x})} \lambda(\mathbf{z}) |\theta(\mathbf{z}) - y| \tag{13}
$$

The conditions (assumptions) that we need for $c(\theta) \leq$ the LHS of (13) is discussed in the appendix. Now, we will continue with

$$
c(\theta) \leq \frac{1}{n} \sum_{(\mathbf{x}, y) \in (\mathbf{X}, \mathbf{Y})} \max_{\mathbf{z} \in \mathcal{Q}(\mathbf{x})} \lambda(\mathbf{z}) |\theta(\mathbf{z}) - y| \tag{14}
$$

When (14) holds, replacing $|\theta(\mathbf{z}) - y|$ with a generic loss $\ell(\theta(\mathbf{z}), y)$ and minimizing it is another direction of learning robust models, which corresponds to distributionally robust optimization (DRO) [Ben-Tal et al., 2013, Duchi et al., 2021].

Further, depends on implementations of $\lambda(\mathbf{x})$, DRO has been implemented with different concrete solutions, sometimes with structural assumptions [Hu et al., 2018], such as

- Adversarially reweighted learning (ARL) [Lahoti et al., 2020] uses another model $\phi : \mathcal{X} \times \mathcal{Y} \rightarrow [0, 1]$ to identify samples with misaligned features that cause high losses of model $\theta$ and defines

$$
\lambda(\mathbf{x}) = 1 + |(\mathbf{X}, \mathbf{Y})| \cdot \frac{\phi(\mathbf{x})}{\sum_{(\mathbf{x}, y) \in (\mathbf{X}, \mathbf{Y})} \phi(\mathbf{x})}
$$

- Learning from failures (LFF) [Nam et al., 2020] also trains another model $\phi$ by amplifying its early-stage predictions and defines

$$
\lambda(\mathbf{x}) = \frac{\ell(\phi(\mathbf{x}), y)}{\ell(\phi(\mathbf{x}), y) + \ell(\theta(\mathbf{x}), y)} \tag{15}
$$

- Group DRO [Sagawa* et al., 2020] assumes the availability of the structural partition of the samples, and defines the weight of samples at partition $\mathbf{g}$ as

$$
\lambda(\mathbf{x}) = \frac{\exp(\ell(\theta(\mathbf{x}), y)))}{\sum_{(\mathbf{z}, y) \in (\mathbf{X}, \mathbf{Y})_{\mathbf{g}}} \exp(\ell(\theta(\mathbf{z}), y)))}, \tag{16}
$$

if $(\mathbf{x}, y) \in (\mathbf{X}, \mathbf{Y})_{\mathbf{g}}$, samples of partition $\mathbf{g}$

These discussions are expanded in the appendix.

## 4.2 REGULARIZING THE HYPOTHESIS SPACE

Connecting our theory to the other main thread is little bit tricky, as we need to extend the model to an encoder/decoder structure, where we use $e_\theta$ and $d_\theta$ to denote them respectively. Thus, by definition of classification models, we have $\theta(\mathbf{x}) = d_\theta(e_\theta(\mathbf{x}))$. Further, we define $f'_m$ as the equivalent of $f_m$ with the only difference is that $f'_m$ operates on the representations $e_\theta(\mathbf{x})$. With the setup, optimizing the empirical loss and $c(\theta)$ leads to (details in the appendix):

$$\min_{d_\theta, e_\theta} \frac{1}{n} \sum_{(\mathbf{x}, y) \in (\mathbf{X}, \mathbf{Y})} \ell(d_\theta(e_\theta(\mathbf{x})), y) - \ell(f'_m(e_\theta(\mathbf{x})), y), \tag{17}$$

which is highly related to methods used to learn auxiliary-annotation-invariant representations, and the most popular example of these methods is probably DANN [Ganin et al., 2016].

Then, the question left is how to get $f'_m$. We can design a specific architecture given the prior knowledge of the data, then $f'_m$ can be directly estimated through

$$\min_{f'_m} \frac{1}{n} \sum_{(\mathbf{x}, y) \in (\mathbf{X}, \mathbf{Y})} \ell(f'_m(e_\theta(\mathbf{x})), y), \tag{18}$$

which connects to several methods in Section 2, such as [Wang et al., 2019a, Bahng et al., 2019]. Alternatively, we can estimate $f'_m$ with additional annotations (*e.g.*, domain ids, batch ids *etc*), then we can estimate the model by (with $\mathbf{t}$ denoting the additional annotation)

$$\min_{f'_m} \frac{1}{n} \sum_{(\mathbf{x}, \mathbf{t}) \in (\mathbf{X}, \mathbf{T})} \ell(f'_m(e_\theta(\mathbf{x})), \mathbf{t}), \tag{19}$$

which connects to methods in domain adaptation literature such as [Ganin et al., 2016, Li et al., 2018].

## 4.3 A NEW HEURISTIC: WORST-CASE TRAINING WITH REGULARIZED HYPOTHESIS SPACE

Our analysis showed that optimizing for small $c(\theta)$ naturally connects to one of the two mainstream families of methods used to train robust models in the literature, which naturally inspires us to invent a new method by combining these two directions. The intuition behind this design rationale is to incorporate the empirical strength of each of these methods together by directly combining the major components of these methods.

Therefore, we introduce a new heuristic that combines the worst-case training (11) and the regularization method (17) and (19), for which, whether the samples are originally from $(\mathbf{X}, \mathbf{Y})$ or generated along the training will serve as the additional annotation $\mathbf{t}$.

---

**Algorithm 1:** worst-case training with regularized hypothesis space

**Result:** $\theta^I$
**Input:** total iterations $I$, $(\mathbf{X}, \mathbf{Y})$;
initialize $\theta^{(0)}$, $i = 1$;
**while** $i \leq I$ **do**
    **for** *sample* $(\mathbf{x}, y)$ **do**
        assign additional label $\mathbf{t_x} = 0$ for $\mathbf{x}$;
        sample $\mathbf{z} \in Q(\mathbf{x})$ that maximizes $\ell(\theta(\mathbf{x}), y)$;
        assign additional label $\mathbf{t_z} = 1$ for $\mathbf{z}$;
        update $f'_m$ with (19);
        update $\theta$ with (17);
        update $\theta$ with $\mathbf{z}$ with the equivalence of (17) as
          $\min_{d_\theta, e_\theta} \ell(d_\theta(e_\theta(\mathbf{z})), y) - \ell(f'_m(e_\theta(\mathbf{z})), y)$,
    **end**
**end**

---

In particular, our heuristic is illustrated with Algorithm 1. In practice, we will also introduce a hyperparamter to balance the two losses in (17).

## 5 EXPERIMENTS

We presented the theory supporting experiments in Appendix, and discuss performance competing results here.

To test the performance of our new heuristic, we compare our methods on a fairly recent and strong baseline. In particular, we follow the setup of a direct precedent of our work [Bahng et al., 2019] to compare the models for a nine super-class ImageNet classification [Ilyas et al., 2019] with class-balanced strategies. Also, we follow [Bahng et al., 2019] to report standard accuracy, weighted accuracy, a scenario where samples with unusual texture are weighted more, and accuracy over ImageNet-A [Hendrycks et al., 2019b], a collection of failure cases for most ImageNet trained models. Additionally, we also report the performance over ImageNet-Sketch [Wang et al., 2019a], an independently collected ImageNet test set with only sketch images.

We test our method with the pipeline made available by [Bahng et al., 2019], and we compare with the vanilla network, and several methods that are designed to reduce the texture bias: including StylisedIN (SN) [Geirhos et al., 2019], LearnedMixin (LM) [Clark et al., 2019], RUBi [Cadene et al., 2019], and ReBias [Bahng et al., 2019], several other baselines proved effective in learning robust models, such as Mix-up [Zhang et al., 2017], Cutout [DeVries and Taylor, 2017], AugMix Hendrycks et al. [2019a], In addition, we compared our worst-case training (WT), regularization (Reg), and the introduced heuristic (WR). For our methods, we follow the observations in [Wang et al., 2020] suggesting the relationship between frequency-based perturbation and the model's performance, and design the augmentation of frequency-based perturbation with different radii.

| | Vanilla | SN | LM | RUBi | ReBias | Mixup | Cutout | AugMix | WT | Reg | WR |
|---|---|---|---|---|---|---|---|---|---|---|---|
| Standard Acc. | 90.80 | 88.40 | 67.90 | 90.50 | 91.90 | 92.50 | 91.20 | 92.90 | 92.50 | 93.10 | **93.30** |
| Weighted Acc. | 88.80 | 86.60 | 65.90 | 88.60 | 90.50 | 91.20 | 90.30 | 91.70 | 91.30 | **92.20** | 92.00 |
| ImageNet-A | 24.90 | 24.60 | 18.80 | 27.70 | 29.60 | 29.10 | 27.30 | **31.50** | 28.50 | 30.00 | 29.60 |
| ImageNet-Sketch | 41.10 | 40.50 | 36.80 | 42.30 | 41.80 | 40.60 | 38.70 | 41.40 | 43.00 | 42.50 | **43.20** |
| average | 61.40 | 60.03 | 47.35 | 62.28 | 63.45 | 63.35 | 61.88 | 64.38 | 63.83 | 64.45 | **64.53** |

Table 1: Results comparison on nine super-class ImageNet classification.

We report the results in Table 1. Our results suggest that, while the augmentation method we used is much simpler than the ones used in AugMix, our empirical results are fairly strong in comparison. With simple perturbation inspired from [Wang et al., 2020], our new heuristic outperforms other methods in average on these four test scenarios.

# 6 DISCUSSION

Before we conclude, we would like to devote a section to discuss several topics more broadly related to this paper.

**Human-aligned machine learning may not be solvable in general without prior knowledge.** Following our notations in this paper, for any two functions $f_1$ and $f_2$, it is human, instead of any statistical properties, that decides whether $f_h = f_1$ or $f_h = f_2$. This remark is a restatement of our motivating example in Figure 1. Our proposed method forgoes the requirement of prior knowledge and is validated empirically on certain benchmark datasets.

**Do all the model's understandings of the data have to be aligned with a human's?** Probably no. As we have discussed in the preceding sections, we agree that there are also scenarios where it is beneficial for models' perception to outperform a human's. For example, we may expect the models to outperform the human vision system when applied to make a scientific discovery at a molecule level. This paper investigates these questions for the scenarios where the alignment is essential.

**In practice, there is probably more than one source of misaligned features.** We aim to contribute a principled understanding of the problem, starting with its basic form. The extension of our analysis to multiple sources of misaligned features is considered a future direction.

**Differences between overfitting and non-human-aligned** A critical difference is that overfitting can typically be observed empirically with a split of train and test datasets, while learning the misaligned features is usually not observed because the misaligned features can be true across the train and test data split.

**Other related works** There is also a proliferation of works that aim to improve the robustness of machine learning methods from a data perspective, such as the methods developed to counter spurious correlations [Vigen, 2015], confound-

ing factors [McDonald, 2014], or dataset bias [Torralba and Efros, 2011]. We believe how our analysis is statistically connected to these topics is also an interesting future direction. Further, there is also an active line of research aiming to align the human and models' perception of data by studying how humans process the images [Kubilius et al., 2019, Marblestone et al., 2016, Nayebi and Ganguli, 2017, Lindsay and Miller, 2018, Li et al., 2019, Dapello et al., 2020].

In addition, discussion of how human annotation will help the models to generalize in non-trivial test scenarios has also been explored. For example, [Ross et al., 2017] built expert annotation into the model to regularize the explanation of the models to counter the model's tendency in learning misaligned features. The study has been extended with multiple follow-ups to introduce human-annotation into the interpretation of the models [Schramowski et al., 2020, Teso and Kersting, 2019, Lertvittayakumjorn et al., 2020], and shows that the human's knowledge will help model's learning the concepts that can generalize in non-i.i.d scenarios.

# 7 CONCLUSION

In this paper, we built upon the importance of learning human-aligned model and studied the generalization properties of a model for the goal of the alignment between the human and the model. We extended the widely-accepted generalization error bound with an additional term for the differences between the human and the model, and this new term relies on how the misaligned features are associated with the label. Optimizing for small empirical loss and small this term will lead to a model that is better aligned to humans. Thus, our analysis naturally offers a set of methods to this problem. Interestingly, these methods are closely connected to the established methods in multiple topics regarding robust machine learning. Finally, by noticing our analysis can link to two mainstream families of methods of learning robust models, we propose a new heuristic of combining them. In a fairly advanced experiment, we demonstrate the empirical strength of our new method.

## Acknowledgement

This work was supported in part by NIH R01GM114311, NIH P30DA035778, NSF IIS1617583, NSF CAREER IIS-2150012 and IIS-2204808.

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
