# OpenReview forum: "Toward Learning Human-aligned Cross-domain Robust Models by Countering Misaligned Features"
_auai.org/UAI/2022/Conference — UAI 2022 Poster_

### Official Review · Reviewer_9qoz · 2022-04-10

**Q2(1) Originality/Novelty:** 1
**Q2(2) Significance/Impact:** 1
**Q2(3) Correctness/Technical Quality:** 3
**Q2(6) Clarity Of Writing:** 4
**Q6 Overall Score:** 6
**Q8 Confidence In Your Score:** 5

**Q1 Summary And Contributions:**

The manuscript attempts to analyze techniques to tackle the Out-of-Distribution(OOD) problem. The authors refer to the spurious features as the 'misaligned' features and the invariant features as the 'human-aligned' features. The authors claim to improve on past methods' performance by curating and combining a set of methods inspired by prior work.

**Q2 Assessment Of The Paper:**

More detailed information regarding each of these aspects is given below:

**Q2(4) Quality Of Experiments (Optional):**

1: Poor: The experimental evaluation is flawed or the results fail to adequately support the main claims.

**Q2(5) Reproducibility:**

1: Poor: Key details (e.g., proof sketches, experimental setup) are incomplete/unclear, or key resources (e.g., proofs, code, data) are unavailable.

**Q3 Main Strengths:**

The manuscript is written in clear and concise prose. Most related work is covered. The hypothesis, assumptions, and claims are formally stated.  The manuscript is neatly divided into sensible sections.

**Q4 Main Weakness:**

The authors have used a mixture of different methods from prior works to form a hypothesis. The manuscript should use the same vocabulary as the prior work where possible. Most OOD datasets are annotated by humans, so using 'human-aligned' and 'misaligned' is not necessary and only results in the mischaracterization of the task.

The main problem with the manuscript is the lack of experiments, experimentation details, and connection to the other sections. The authors only use variants of the ImageNet dataset to support their claims. But the common practice in this area is to use datasets across several different tasks. The authors mention that they use the experimental setup from a prior debiasing work. However, the problem they are trying to solve is different. The experimental setup, model selection, and hyperparameters should be stated in detail.
The model should be compared with domain-generalization benchmark models commonly used in prior work rather than texture bias models.

**Q5 Detailed Comments To The Authors:**

A lot of work is needed in the experiments section. Please try to incorporate the SoTA models from DomainBed[1] or Wilds[2]. To properly show that a model is learning the invariant or 'human-aligned' features, the model selection has to be based on the training environment. However, I was not able to see any details about this in the manuscript. It also appears that there is some attempt to link debiasing in this work because of the experimental choices. Even though debiasing might be associated with this problem, it is different. It would be interesting to see if the approach works on non-image datasets as well.

[1] Gulrajani, Ishaan, and David Lopez-Paz. "In search of lost domain generalization." _arXiv preprint arXiv:2007.01434_ (2020).
[2] Koh, Pang Wei, et al. "Wilds: A benchmark of in-the-wild distribution shifts." _International Conference on Machine Learning_. PMLR, 2021

**Q7 Justification For Your Score:**

My overall score is based on the usefulness of the work. If the author's method needs to be evaluated properly to judge its usefulness. The experimental setup and details weighed heavily towards the score. Even though the manuscript is well written, the writing was given a low score since the writing must be backed up with something concrete.


**Q9 Complying With Reviewing Instructions:**

1: Yes.

---

### Official Review · Reviewer_4ezZ · 2022-04-12

**Q2(1) Originality/Novelty:** 3
**Q2(2) Significance/Impact:** 2
**Q2(3) Correctness/Technical Quality:** 3
**Q2(6) Clarity Of Writing:** 3
**Q6 Overall Score:** 5
**Q8 Confidence In Your Score:** 2

**Q1 Summary And Contributions:**

This work is motivated by the perspective that the reason behind the accuracy drop is the reliance of models on the features that are not aligned well with how a data annotator considers similar across these two datasets.

Then the authors extend the conventional generalization error bound to a new one for this setup with the knowledge of how the misaligned features are associated with the label and a corresponding solution.

**Q2 Assessment Of The Paper:**

More detailed information regarding each of these aspects is given below:

**Q2(4) Quality Of Experiments (Optional):**

2: Fair: The experimental evaluation is weak: important baselines are missing, or the results do not adequately support the main claims.

**Q2(5) Reproducibility:**

2: Fair: Key resources (e.g., proofs, code, data) are unavailable but key details (e.g., proof sketches, experimental setup) are sufficiently well-described for an expert to confidently reproduce the main results.

**Q3 Main Strengths:**

The new perspective of understanding why the accuracy drops on another distribution is interesting.
The theoretical analysis is sound.
It is interesting to see that the methods our analysis leads to are closely connected to the established methods in multiple topics regarding robust machine learning.
The discussion part is insightful and makes this work complete.

**Q4 Main Weakness:**

The experiment part is weak. There are so many benchmark datasets for domain shift, which should be employed to verify the proposed method. There are also so many baselines that should be compared.

Besides, the designed experiment is not strongly associated with the proposed theory. More details and explanations are needed.

**Q5 Detailed Comments To The Authors:**

The motivation is strong. The theoretical analysis is sufficient. The experiment could be more adequate.

**Q7 Justification For Your Score:**

I think the technique of this work is solid but the experiment part can be enriched. Thus I arrived at 6.

**Q9 Complying With Reviewing Instructions:**

1: Yes.

---

### Official Review · Reviewer_JrAX · 2022-04-13

**Q2(1) Originality/Novelty:** 3
**Q2(2) Significance/Impact:** 3
**Q2(3) Correctness/Technical Quality:** 2
**Q2(6) Clarity Of Writing:** 2
**Q6 Overall Score:** 5
**Q8 Confidence In Your Score:** 4

**Q1 Summary And Contributions:**

The authors study the problem of models relying on irrelevant features.  Their main contributions are: 1) a new generalization error bound for this setting; 2) a number of techniques for aligning the model derived from this bound; and 3) a numerical comparison of alternative techniques.

**Q2 Assessment Of The Paper:**

More detailed information regarding each of these aspects is given below:

**Q2(4) Quality Of Experiments (Optional):**

2: Fair: The experimental evaluation is weak: important baselines are missing, or the results do not adequately support the main claims.

**Q2(5) Reproducibility:**

3: Good: Key resources (e.g., proofs, code, data) are available and key details (e.g., proofs, experimental setup) are sufficiently well-described for competent researchers to confidently reproduce the main results.

**Q3 Main Strengths:**

1. The topic is very relevant for UAI and AI at large and the contribution will be of interest to specialists working on model alignment and debugging, and synergizes well with existing works.  A theoretical understanding of why feature alignment methods work is useful and can serve as a stepping stone for further research.
2. The paper is well structured and generally readable.  Many ideas are expressed clearly, but some important concepts (see below) are not.  English and notation are a bit contrived in places.  Some parts feel hastily written (sections 4.* come to mind).
3. The literature review is well done.  There is more related work that is not covered here, but the authors were quite exhaustive already.  The numerical experiments are not described in great detail in the main text, but they are not particularly important in the economy of the manuscript.

All in all, a good paper, but some elements need to be clarified during rebuttal before I can safely recommend acceptance.

**Q4 Main Weakness:**

The domain of the optimization operation in Eq. 5 is not obvious and should be formalized explicitly.  (The explanation right after Eq 5 is not entirely obvious to me.)

Why is y bold?  Is it a vector or a scalar?  It appears to be a vector in e.g. eq 17, but then in eq 13 and elsewhere the difference between theta(z) and y is measured using a regular absolute value (not a proper norm between vectors). This is very confusing.

Section 4.2 is particularly hard to parse.  Decoders generally map from latent space back  to input space, so how can theta(x) = d_theta(e_theta(x))?

**Q5 Detailed Comments To The Authors:**

- p 3: presumably \mathcal{X} should be a subset of R^p and \mathcal{Y} should be exactly {0, 1}?
- p 3: the description of various assumptions A1 should be converted to natural language.
- p 4: "A2 can be interpreted as a regulation" -> I had trouble parsing this sentence, I'd replace it with "moreover, A2 ensures that f_m exists for P_s" or similar.
- p 4: "we consider the differences lie in the features they use" -> we focus on the case where the difference lies in the features they use.
- p 4: "we believe the A in both LHS and RHS" -> we imply that A is the same in both LH and RHS.
- p 4: "d(theta, fh, x) d(theta, fm , x) = 0" would be easier to parse if rewritten as "d(theta, fh , x) = 0 or d(theta, fm , x) = 0".
- p 4: how is y determined in Eq. 5?  is it y = f(x)?  how are the features of z outside of A(f, x) picked?  wouldn't it make more sense to use the same signature as for d (I mean r(theta, f, x))?  this would be more symmetric. Please consider correcting the notation accordingly.
- p 4: "can no longer indicate" -> is not representative of?
- A5: what is f_d?
- p 5: "for an arbitrary target dataset (X, Y)_{P_t}, no samples out of which can be predicted correctly by learning f_m (a situation likely to occur for arbitrary datasets)".  It is not that common, right?  Even if the machine relies on confounded features for some class, it typically manages to work well for unconfounded inputs.  I'm not sure what the message is here.
- p 6: please clarify that (X, Y) is drawn from P_s.
- p 6: the adversarial training -> remove the.
- p 6: lambda(z) is used without being defined.  it is implicitly introduced only a few paragraphs after its first mention.
- p 7: extra z after full stop.
- please check the bibliography - several entries are badly formatted, e.g., LIME and Sagawa and Koh.

**Q7 Justification For Your Score:**

A relevant and interesting paper that can be accepted if the authors manage to clarify the main points.  The weaknesses of this paper are mostly about clarity and notation.  In this case, since this paper is chiefly theoretical, clarity and notation are important.

**Q9 Complying With Reviewing Instructions:**

1: Yes.

---

### Official Review · Reviewer_QRZF · 2022-04-24

**Q2(1) Originality/Novelty:** 3
**Q2(2) Significance/Impact:** 3
**Q2(3) Correctness/Technical Quality:** 3
**Q2(6) Clarity Of Writing:** 3
**Q6 Overall Score:** 6
**Q8 Confidence In Your Score:** 3

**Q1 Summary And Contributions:**

The authors study the problem of ‘misaligned’ features, the case where the features used for automatic prediction differ from those used by humans for ‘natural’ prediction — causing potentially catastrophic test data outcomes. They specifically: (1) provide a new generalization bound based on the difference of the ‘active’ feature set used by the algorithm and human, (2) use this analysis to motivate a new heuristic, and (3) validate this heuristic on benchmark datasets against solid baselines.

**Q2 Assessment Of The Paper:**

More detailed information regarding each of these aspects is given below:

**Q2(4) Quality Of Experiments (Optional):**

2: Fair: The experimental evaluation is weak: important baselines are missing, or the results do not adequately support the main claims.

**Q2(5) Reproducibility:**

3: Good: Key resources (e.g., proofs, code, data) are available and key details (e.g., proofs, experimental setup) are sufficiently well-described for competent researchers to confidently reproduce the main results.

**Q3 Main Strengths:**

- This is a problem of increased recent interest (especially in deep learning) and the theory is well-contextualized regarding recent methods that robustify through either perturbation to reduce incidence of (consistent) spurious features, domain-specific hypothesis space regularization, and domain conditioning as an input. This work makes some progress regarding a formal analysis that supports these different paradigms.
- The central formulation (A2) is conceptually sensible. While f_m for all x in P being a somewhat strong assumption, it isn’t completely unreasonable and does facilitate making some progress on this problem
- The provided empirical comparisons are relative to multiple strong baselines and performs competitive with SotA for this setting.
- The discussion in section 6 does point out conceptual limitations of this problem in general and the appendices helped answer some of my questions. Once the appendices are considered, this is a notably well-written paper.


**Q4 Main Weakness:**

- The “f_m for all x in P” is strong assumption; it seems that this could have a bounding factor to improve the analysis.
- Knowledge of f_m is also a strong assumption as knowledge of the active set can lead to direct hypothesis regularization methods from a practical perspective; while these assumptions are enabling, once made, (6) is fairly straightforward.
- The reliance on the active feature set implies feature independence in the active set comparison; there can be correlations and interactions that would still align the features to some degree.
- Use of mixing hyperparameter isn’t clear.
- The discussion of empirical results in the main paper is missing a ‘deep dive’ case to contrast differences. The discussion in the appendices is ‘appendix-appropriate’ meaning the paper never really has a strong discussion of the empirical results. The existing discussion is more about conceptual limitations and where to look next to advance a theory for this problem.


**Q5 Detailed Comments To The Authors:**

Overall, I like the general formulation and have thought about it more than most papers I review — and think the paper is well-motivated and written. To summarize the strengths and weaknesses from previous sections, from a theoretical perspective the strengths are that the proposed theory supports many existing solutions to this problem and results in a new, more straightforward solutions that performs competitively while the primary weakness is that the assumptions are a bit stronger than desired and I think can actually be loosened a bit with bounding divergences (but this could be a direction for future work). The connections to domain adaptation and adversarial features help give some intuitive contextualization regarding the proposed analytical formulation and the new heuristic is simpler and more understandable than existing more problem/domain-specific work. From an empirical perspective, the main paper pretty much shows marginally better performance against recent baselines (so there is not a breakthrough in this regard). However, I would like more discussion of the experiments — even with the appendices, I had to speculate regarding explanations for different performance results between algorithms.


**Q7 Justification For Your Score:**

This is a direction of increased recent interest and the theoretical formulation both supports previously proposed methods and emits a new approach that performs well empirically. My primary concern regarding this paper is that the empirical results need a deeper dive to really understand why we are seeing the relative methodological performance. My secondary concern is that the theory advances understanding, but some of the assumptions can be loosened and result in a more complete theory.


**Q9 Complying With Reviewing Instructions:**

1: Yes.

---

### Decision · Program_Chairs · 2022-05-15

**Decision:**

Accept (Poster)

**Comment:**

Meta Review: This paper receives generally positive ratings. After rebuttal, all reviewers recommend acceptance. One reviewer has concerns that authors should also include domain generalization settings and baselines. AC agrees with this but also understands that it is impossible for one paper to incorporate every piece of OOD such as: Cross-domain/Domain Transfer/Domain Adaptation/Domain Generalization/Few-shot/Zero-Shot/Open-set, etc. As well as the most recent WILDS benchmarks and DomainBed settings, So, AC suggests the authors position this paper topic more clearly in the introduction and experiments. AC recommends acceptance.